# Impact of Ants on the Order Composition of Canopy Arthropod Communities in Temperate and Tropical Forests

**DOI:** 10.3390/ani15131914

**Published:** 2025-06-28

**Authors:** Andreas Floren, Tobias Müller

**Affiliations:** 1Department of Bioinformatics, Biocentre, University of Würzburg, Am Hubland, 97074 Würzburg, Germany; tobias.mueller@uni-wuerzburg.de; 2Department of Animal Ecology and Tropical Biology, Biocentre, University of Würzburg, Am Hubland, 97074 Würzburg, Germany

**Keywords:** community composition, count data, seasonality, ant predation, forest disturbance, regeneration, canopy fogging

## Abstract

Canopy ants influence the diversity and composition of arthropod communities in tropical primary rainforests and temperate forests. In the rainforests, ants are active year-round, suppressing the dominance of any single species and preventing pest outbreaks. In contrast, seasonality restricts ant diversity and their impact on arthropods in temperate forests. However, this changes when ground-dwelling ants enter the canopy to prey on arthropods. Nevertheless, significantly more arthropods were collected from trees with high ant density compared to trees with low or no ant abundance in tropical and temperate forests. This may be related to the consistently high availability of aphid honeydew in temperate forests, but honeydew is less abundant in rainforests and cannot explain the high arthropod abundance observed here. In secondary forests, the functional influence of ants is lost, resulting in fundamental community changes. Our results highlight that forest regeneration requires recolonization from a primary forest and takes over 50 years to fulfill primary forest functions.

## 1. Introduction

Ants are among the most abundant arthropods and are considered ecosystem engineers due to their high ecological impact [1]. They are particularly abundant and extremely diverse in tropical lowland rainforests where they inhabit all trees [2,3,4]. They dominate the canopy, where they forage continuously and exert high selection pressure on the canopy associated fauna [5]. However, little is known about how ants influence community composition at the ordinal level, which also determines the composition of species communities. This can be illustrated by the example of honeydew-producing hemipterans, which play a key role in insect–plant interactions, e.g., through trophobiosis or kairomones, which indicate the location of hosts or prey [6,7].

To investigate the impact of ants, we sampled arthropod communities as thoroughly as possible using insecticidal knock-down fogging in temperate forests and lowland rainforests in Sabah, Malaysia [8]. From all trees in the rainforest, we selected those with high ant abundance and compared them with trees from which significantly fewer ants had been collected. These comparisons were made in the dry and the wet season to assess how strong seasonality influences the abundance distribution of arthropods [9,10]. Anthropogenic disturbance is another factor that strongly alters the diversity and structure of arthropod communities [11,12,13], but only in a few countries, such as Malaysia, is it still possible to calibrate the effects on primary forests, which is why we extended our research to secondary forests.

Ants are also regularly found in the trees of the temperate forests of Central Europe, but in lower diversity and abundance, resulting in a less pronounced influence on arthropod communities [14]. However, this changes when ground-nesting ants of the genus *Formica* are present in a forest and thousands of ants enter the trees to forage. An outstanding example is *F. polyctena*, an extremely effective predator that obtains a substantial proportion of its protein-rich food from canopy arthropods, particularly during mass outbreaks of pest species [15]. Although this has been extensively studied, we are not aware of any investigation that has examined the effects of *F. polyctena* (or other members of the *Formica* group) on the composition of canopy arthropod communities. As in the tropics, sampling was repeated in both early and late seasons, when arthropod abundance declines, and prey availability is severely reduced. As all primary forests in Central Europe have been destroyed or modified to meet human needs, it is not possible to directly assess the impact of anthropogenic disturbance on canopy diversity.

In summary, we ask how the ecological role of canopy ants differs between tropical and temperate forests, and to what extent seasonality influences these patterns in both biomes. Although seasonality is less pronounced in the tropics than at temperate latitudes, there is limited information on how this affects community diversity and ecosystem function [10]. Finally, we investigate how ant communities in secondary forests change over time and how these shifts impact the canopy fauna.

## 2. Materials and Methods

### 2.1. Study Sites

#### 2.1.1. Temperate Forests in Central Europe

Research was conducted in two xerothermic oak forests (*Quercus robur*) near Münnerstadt (Germany, 50°12′56.7″ N 10°15′59.9″ E) in June and August 1998. The two forests were located one kilometer apart. Numerous *F. polyctena* nests were present only at one of the forest sites. Elevation was 330 m above sea level, and the mean annual precipitation was 465 mm. The average age of the oaks was 120 years. Both forests were similar in structure and classified as *Querceto-Fagetea*. They were formerly managed as coppice with standards but had been transformed into mixed forests decades earlier. *Q. robur* dominated at both sites, while *Carpinus betulus*, *Tilia cordata*, *Sorbus torminalis*, *Acer platanoides*, and *Fagus sylvatica* were regularly found in lower numbers. The number of *Formica polyctena* ants was estimated at 27 million individuals [16], thousands of which entered the canopy in large trails to forage. In June, seven fully grown oak trees—with and without *F. polyctena*—were fogged in each forest. In August, the same sampling procedure was followed, but only five trees with low ant abundance could be fogged (Appendix A). All newly selected trees were far enough away (>100 m) from the previously fogged trees and had not been affected by the June fogging.

#### 2.1.2. Tropical Primary Lowland Rain Forests in Southeast Asia

Fieldwork in the tropics was conducted in the lowland rainforests of Mount Kinabalu National Park in Sabah, Malaysia, on Borneo. Forests grew at an elevation of about 700 m above sea level [17]. The climate is tropical and hot, with temperatures ranging from 25 °C in January to 27 °C in September. Rainfall averages 3700 mm per year and varies seasonally, with 200–250 mm per month during the dry season and 350–400 mm in the wet season. There is no strict separation between seasons, as it typically rains every day. Humidity remains around 80% throughout the year. The National Park supports an estimated 5000 to 6000 plant species within the 700 sq. km area, including about 1000 tree species [18,19]. We followed a similar experimental approach, focusing on trees of the genus *Aporosa* (*A. lagenocarpa* and *A. subcaudata*; Phyllanthaceae), the only tree species found in larger numbers. The mean tree height was 24 m. Ants inhabited all trees, as evidenced by exhaustive hand sampling and fogging. In the rainforest, we examined the effect of ants on arthropod communities by comparing trees with high ant densities to trees where ant abundance was low (Appendix A). These included 14 trees with more than 3000 individuals (“Ant-trees”) and 18 trees with fewer than 500 individuals (“LowAnt-trees”). Seven trees with low ant abundance and nine with high ant abundance were fogged during the dry season (end of January and February). In the wet season (September to November), the respective numbers were eleven and five. Foggings were conducted in 1993 and 1996.

#### 2.1.3. Comparison of Tropical Primary and Disturbed Rain Forests in Southeast Asia

An extended dataset was used to analyze the effects of anthropogenic disturbance on ant communities and the associated arthropod fauna. The primary forest was represented by 35 *Aporosa* trees plus 6 *Xanthophyllum affine* (Polygalaceae), resulting in a total of 41 trees. Six regenerating secondary forest types, varying in age and distance from each other, were selected to analyze the effects of anthropogenic disturbances. We distinguished three gradient forests that had been left to regenerate for 5, 15, and 40 years and which merged with and eventually bordered primary forests. In the 5-year-old site, the common pioneer tree *Melochia umbellata* (Sterculiaceae) was dominant, while *Vitex pinnata* (Verbenaceae) was abundant in the other two sites. From each gradient forest site, 9, 13, and 11 trees were fogged, respectively. The other three forests were 10, 20, and 50 years old and isolated several kilometers from each other. All secondary forests had been cleared mostly for agriculture and were left to regenerate. All had only a single canopy layer. They were located at least 15 km from the primary forests of the Crocker Range National Park. In all three forests, the pioneer tree *Melanolepis* sp. (Euphorbiaceae) was common and selected for fogging. From each isolated forest, 9, 6, and 8 trees were fogged. Fogging was performed in February and March 1997 and 1998. More specific information for each site can be found in [20].

### 2.2. Canopy Fogging and Order Sorting

Canopy fogging effectively collects ectophytic arboreal arthropods in a quantitative and tree-specific way [8]. Natural pyrethrum in a concentration of less than 1% was used as an insecticide. It is highly specific to arthropods and destroyed in direct sunlight within hours without leaving persistent substances in the trees. Natural pyrethrum causes uncoordinated movements of the affected arthropods, which eventually fall from the leaves or branches into the collection trays, explaining its high efficiency. Collection sheets were installed beneath each tree, covering at least 80% of the crown projection area. By precise positioning the collection sheets, arthropods were excluded from neighboring trees. Foggings were conducted early in the morning when there was little air movement. Depending on the weather, the fogging of an individual tree took only a few minutes in the field. All arthropods that dropped into the collecting sheets within two hours after fogging were collected and conserved in 80% ethanol, and taxa were sorted at least at the order level. Within the order Hemiptera, we distinguished between the suborders Heteroptera (true bugs) and Homoptera (plant suckers) due to their different life habits. Due to the dominance of Lachnidae (Sternorrhyncha, Aphidoidea, Homoptera) in the temperate forests, they were also treated separately. Formicidae (Hymenoptera) were treated as major taxa according to their numerical dominance. Specimens of orders represented by only a few individuals were grouped as “Others”.

### 2.3. Statistics

Statistical analyses were performed using the software R version 4.3.2 (R Core Team 2024) and the packages of the Bioconductor project [21]. We used a pyramid plot as implemented in the ggplot2 package [22] to visualize the distribution of the major arthropod groups from trees fogged in tropical and temperate forests. This descriptive overview plot is based on 1284 foggings in Europe (1013 foggings) and Malaysia (271 foggings) over the last 28 years. A correspondence analysis (CA), as implemented in the vegan R-package (Version: 2.6.10) [23], was performed to visualize differences in community structure between arthropod communities for both tropical and temperate forests. To better illustrate the differences between the trees, we also plotted the convex hulls for each treatment. To disentangle changes in community structure from changes in absolute abundance, we analyzed the proportional composition of arthropod groups using generalized additive models (GAMs) with quasibinomial error structure and the corresponding count data by negative binomial regression. This dual approach enabled us to assess both compositional shifts in the arthropod communities and the population-level dynamics in terms of abundance. Proportional differences between taxa were modeled by logistic regression within the GAM framework of the R package ‘mgcv’ (Version: 1.9.3) [24]. We modeled the taxon proportion via Taxon ~ FacAnts*Season, using the quasibinomial family, when overdispersion was detected and where “FacAnts” contrasts high and low ant abundance. The factor “Season” refers to the dry and wet season in the rainforest and to early (June) and late summer (August) in the temperate forests. For the tropical forests, the factor “Year” was implemented as random factor in the model. The marginal effects of the interaction term Ants*Season was visualized as implemented in the R package sjPlot (Version: 2.8.17) [25]. The abundance distribution was modeled as: Counts per group ~ FacAnts*Season + Sheet using the negative binomial family. The number of arthropods was also adjusted with the factor “Gbh” (girth in breast height) but was not found to be significant and was therefore excluded from the model. The factor “Sheet” quantifies the number of collecting sheet for each fogged tree. A non-metric multidimensional scaling (NMDS) was used to visualize Jaccard beta diversity between ant communities as implemented in the vegan R-package [23]. NMDS was used on the Euclidian distances of all trees and the scores of the first two axes were included in the following PERMANOVA model to adjust for spatial autocorrelation: Community ~ DistType + PCoA1 + PCoA2 using 999 permutations (where “DistType” models the forest types with seven levels). For an adequate stress value, we choose k = 3 dimensions. To better illustrate the differences between tree species, we plotted the convex hulls on trees for all comparisons (trees with and without dominant ants in two seasons). The function pairwise.adonis was used to calculate post hoc beta diversity comparisons between tree species [26]. Differences in community structure were visualized by rank abundance curves. *p*-values were considered weakly significant if *p* < 0.05 (*), significant if *p* < 0.01 (**), and highly significant if *p* < 0.001 (***). All *p*-values were adjusted for multiple testing using the method of Benjamini and Hochberg [27].

## 3. Results

To provide a comprehensive overview of taxon composition in both temperate and tropical forests, we constructed a pyramid plot based on the analysis of 1284 fogged trees from Europe and Malaysia, visualizing the distribution of arthropods across these ecosystems (Figure 1A). Arthropod communities in tropical and temperate forest trees were strikingly different in order composition. The high dominance of ants in tropical forests was particularly striking, whereas in temperate forests, ants represented only 4% of the canopy arthropods. Diptera and Coleoptera were also present in high abundance. While aphids were absent in rainforests, they were found in high numbers in temperate forests, along with Homoptera, Heteroptera, Psocoptera, and Lepidoptera (mainly caterpillars). When ants were excluded from the data (Figure 1B), most groups showed a proportional increase in abundance, most notably in Diptera and Hymenoptera, while Coleoptera, Araneae, and Orthoptera increased significantly in the tropical forests. These changes suggest the effects of ants on order composition.

The analyses of the effects of dominant ants on arthropod communities in rainforests are based on a comparison of trees with high ant densities (>3000 workers) and low densities (<500 workers). In total, 115,604 arthropods were included in this analysis. During the dry season, ants represented, on average, 46% of all individuals in the “Ant-trees” and 25% in “LowAnt-trees”. The respective numbers for the wet season were 58% and 30% (Table 1). Despite the differences in the number of foggings, an association between ant abundance and arthropod abundance can be inferred, which was later confirmed through modeling. As shown by the marginal interaction plots, overall, more arthropod individuals were collected from trees with high ant dominance in both seasons, with 4.7 times more arthropods in the dry season and 3.9 times more in the wet season (Appendix A). This effect was most pronounced for Diptera, Coleoptera, parasitic Hymenoptera, and Lepidoptera.

Community structure was different in the temperate forests. When *F. polyctena* was present, they dominated all trees and provided 23,800 individuals overall (37.1%), followed by the Lachnidae, for which 8107 individuals (12.6%) were collected. In August, the number of ants increased to 63,833 (66.5%), while Lachnidae reached 1396 individuals (14.5%). As in the tropical rainforests, arthropod abundance was highest on the “Ant-trees” (even without considering ant numbers). In June, 2.9 times more arthropods were fogged from the ‘Ant-trees’, and in August, 10.5 times more were fogged, (Appendix A), which was mainly due to the increase in Lachnidae. Diptera, Hymenoptera, Araneae, und Lepidoptera were always collected in higher numbers compared to the ‘LowAnt-trees’. Homoptera, Heteroptera, and Lepidoptera were mostly collected as juveniles in June but as adults in August.

In temperate forests, arthropod communities were clearly separated depending on whether trees were ant-dominated or not (Figure 2). In a correspondence analysis, the first axis represents the ant effect (70.4% explained variance) and the second axis the seasonal effect (14.17% explained variance). Arthropod abundance was greatly reduced in August, where *F. polyctena* and the honeydew-excreting Lachnidae were grouped together, indicating a close trophic association. As in temperate forests, community composition in the tropics differed between ant-dominated trees and those with low ant abundance. This separation was stronger in the wet than in the dry season. The large convex hulls and partially overlapping clusters show high variation in arthropod community composition, suggesting weaker seasonal effects compared to temperate forests. Formicidae were not associated with potential honeydew-producing Hemiptera in the rainforest.

We modeled how proportions of the major arthropod groups changed under ant dominance and seasonality (Figure 3). In the temperate forests, arthropod groups differed significantly under *F. polyctena* dominance. In June, relative proportions of parasitic Hymenoptera, Coleoptera, Homoptera and Psocoptera were significantly lower on trees with *F. polyctena* than on trees without *F. polyctena*. This was also true for the Lachnidae, although the association was only weakly significant due to the high variability of aphid numbers on trees without ants. Most arthropod groups differed during the season, including a significant marginal interaction term (Ants*Season) for Diptera (*p* < 0.01), with reversed proportions, and for Heteroptera (*p* < 0.05). The situation in the tropical rainforests was more uniform but showed a weak ‘ant effect’ for Diptera and Homoptera and weak seasonal effects for Araneae and Psocoptera.

The functional significance of arthropod groups became clearer when absolute numbers were modeled (Figure 4). In the rainforest, the predicted marginal means revealed that, during the dry season, arthropod individuals increased significantly in most orders—except for Diptera—under ant dominance. Only Homoptera were significantly more abundant in the wet season (*p* < 0.05). We did not find large aggregations of Hemiptera on any tree. In the temperate oak forests, the strongest ant effect was detected in August, most pronounced for the Lachnidae, which were 220.57 times more abundant on trees with *F. polyctena*. Significant increases were also observed for Hymenoptera, Psocoptera, Lepidoptera, Araneae, and Diptera.

The persistence of primary forests in the tropics allows for the calibration of anthropogenic disturbance effects. For the arboreal ants, the differences between forest types are visualized by NMDS, which separates primary forests, gradient forests, and isolated forests (Figure 5A). Only the gradient forests showed a gradual convergence towards primary forest conditions with increasing time of regeneration. The PERMANOVA confirmed highly significant differences in beta diversity between forest types (see Appendix A). Primary forests differed significantly from all other types, except for the 5-year-old gradient forest, which, in turn, was statistically indistinguishable from the 40-year-old gradient forest. The rank abundance curves clearly distinguish primary, gradient, and isolated forests, with ant communities in the isolated forests showing the least pronounced dominance hierarchy (Figure 5B). In contrast, the more even and widespread occupancy across abundance classes in gradient forests reflects a shift toward the dominance hierarchy observed in primary forests. The Venn diagram, based on abundant ant species sampled with more than 400 individuals, illustrates that primary, gradient, and isolated forests harbor distinct ant assemblages, with limited species overlap among forest types.

## 4. Discussion

Ants are crucial key species that maintain biodiversity and ecosystem functioning in both tropical and temperate forests, albeit in different ways. In tropical rainforests, ants are present in all trees. As opportunistic predators, they exert a constant and high level of predation pressure on the arthropod fauna, which remains unaffected by seasonal changes [5]. However, the seasonal variation in arthropod abundance in trees suggests an understudied dynamic, the ecological consequences of which remain unclear [10]. Ant dominance in the canopy has been established since ants began expanding their nesting sites there during the Late Cretaceous/Early Paleogene, around 66–23 million years ago [28]. Evidently, the canopy fauna has adapted very successfully, as reflected by the extraordinarily high diversity of arboreal species, most of which are highly mobile and collected in low abundance [29,30]. Beyond high predation pressure, ants influence ecosystems through a variety of interactions with plants and a wide range of myrmecophilous species across multiple arthropod groups, making them one of the most ecologically influential animal groups [1,31].

In contrast to the ancient primeval forests of Southeast Asia, the temperate deciduous forests of Central Europe only emerged around 8000 years ago and have been almost entirely shaped by human activity over the past 2000 years [32]. Due to strong seasonality and harsh winters, few arboreal species have evolved to nest and develop in the canopy [28,33,34]. As a result, ants make up only a small portion of the tree-associated arthropod fauna and do not play a dominant ecological role [14,35,36]. This changes, however, in the presence of large ground nests of *Formica* ants, which forage intensively in the canopy and thereby assume dominance, similar to what we consistently observed in primary tropical rainforests. *Formica polyctena*, in particular, is a top predator. A large colony can capture approximately 30 kg of insects per year—equivalent to around ten million prey items—and collect an additional 500 kg of honeydew [15]. More surprising was our finding that arthropod abundance increased significantly for ant-dominated trees in both tropical and temperate forests. This effect was independent of the number of collected ants and especially true for preferred ant prey groups such as Psocoptera, Diptera and Lepidoptera. It was also most pronounced in temperate forests in August, when arthropod abundance typically drops sharply due to seasonal dynamics—as seen in trees without *F. polyctena*. In that month, *F. polyctena* and Lachnidae together accounted for more than 80% of all arthropods collected. With the exception of Diptera, Homoptera, and Heteroptera, the abundance of most groups was even higher than in June, when abundance is usually highest.

The unexpected result of greater arthropod abundance on trees with predatory ants raises important questions about the underlying mechanisms. The excess of honeydew secreted by aphids on leaves and branches may best explain this pattern in temperate forests [37]. It is known that honeydew attracts arthropods, but such a strong effect across all arthropod taxa was unexpected. Interestingly, recent studies show that the presence of ants themselves can alter honeydew quality [6,7]. This may indicate that the ants influence this ‘mass effect’ by ensuring the availability of protein-rich food even in August, when arthropod abundance on the “LowAnt-trees” declines. Additionally, ants may benefit from a decline in aphid predators, although we observed an increase in the number of parasitoids on trees with high ant abundance in the fall. In June, when aphids occur in lower numbers and the abundance of arthropods—especially of caterpillars—is highest, this effect is less apparent. High arthropod abundance may also affect community function, which is linked to total abundance [8]. However, this hypothesis can only be evaluated through more detailed studies.

Applying the ‘honeydew hypothesis’ to tropical rainforests is more difficult. Although we found a significant association between ant abundance and hemipterans, there were too few plant-suckers per tree to explain the observed increase in overall arthropod abundance in trees dominated by ants, particularly when compared to the thousands of aphids recorded from the temperate forests. The absence of large aggregations of honeydew-producing hemipterans suggests that honeydew is not the main food source for arboreal ants in primary forests [7,38]. Since extrafloral nectaries are not known to be from *Aporosa* trees, it is reasonable to assume that the ants used other plant exudates. However, the main question remains: How can the significantly higher number of arthropods in the presence of ant dominance be explained? An increase in the abundance of arthropods in all orders on trees with high ant dominance—for example, due to higher niche or food availability—cannot be inferred from our data. Further research, particularly at the species level, is necessary and is currently underway. One hypothesis is that the ants directly attract arthropods, as this happens indirectly through the honeydew in temperate trees. The complexity of chemical communication between ants and other arthropods is evident in the thousands of myrmecophile species, parasites, and predators found in tropical forests [1,39], revealing the complexity of species interactions in primary forests. In disturbed forests, the ant and the associated arthropod communities are exposed to disturbances that disrupt interaction networks, community diversity, and structure, including the chemical communication between species [31,40].

### Disturbance Alters Canopy Ant Communities

In the tropical rain forests, the canopy ant communities clearly illustrate the ecological disruptions caused by anthropogenic disturbance. In all secondary forests studied, community diversity and structure were severely reduced as a result of simpler vegetation structures, a different microclimate, and a decrease in food and nesting resources [10,13,41]. This was accompanied by a change in the ecological role of the canopy ants, which did no longer exert a high predation pressure throughout the canopy [5]. As a result, several arthropod species became highly abundant, aphids and other hemipterans were found in large aggregations, and caterpillars were collected in abundance. In nearly 20 years of research, we have not observed anything similar in primary forests. The presence of multiple nests of the aggressive pioneer ant *Oecophylla smaragdina* was exclusively observed within the secondary forests. Their impact was confined to a limited number of trees, however, indicating that their influence did not extend to large-scale control. This is an important difference compared to primary forests, where ants exert a permanent selection pressure on all animals [1] and it is a striking similarity to the locally limited occurrence and influence of *Formica* species in temperate forests. Decreased ant predation rates were also associated with increased disturbance in Madagascar [40], indicating food web disruption, which often leads to mass outbreaks of pest species and cascading effects on ecological organization [42]. In the tropics and temperate latitudes, predatory ants have been used for pest control, mainly against caterpillars of Lepidoptera and Symphyta, but also against wood-boring bark beetles [15,43].

Our findings underscore the crucial role of primary forests as reservoirs for species recolonization. Only transitional forests with direct proximity to intact primary forests approached similar levels of community structure and diversity in anthropogenically relevant times. Even after 50 years of natural regeneration, the secondary forest differed greatly from the primary forest, suggesting that significantly longer regeneration times are needed to reach primary forest conditions; see also [44]. Our results contrast with studies from Papua New Guinea that report an increase in functional diversity of arboreal ant communities in secondary forests with lower species richness [45]. It was also suggested that arboreal ants may play a comparable ecological function in primary and secondary forests [11]. These results are strikingly similar to those obtained in the old-growth gradient forest, implying that the primary forest in Papua New Guinea is likely an old-growth secondary forest.

In the temperate forests, the analysis of anthropogenic disturbances is limited to the effects of forest management and is therefore difficult to compare—especially with regard to forest regeneration [46,47]. While the arboreal fauna seems to be less affected than in tropical lowland forests, there are indications that the density of *Formica* species in forests has decreased significantly as a result of management, with negative consequences for biodiversity, soil biology and pest control [15,48]. From a conservation perspective, it is even more puzzling that wood ants still play such a minor role in forest management, while the need to protect the high biodiversity and its functions in tropical rainforests is well known but not adequately enforced, e.g., [49].

## 5. Conclusions

In temperate forest trees, ants have little influence on the composition of the associated arthropod communities. This changes in the presence of abundant, ground-dwelling ant species of the genus *Formica*, which affects arthropod communities directly through high predation pressure and indirectly through trophobiosis with honeydew-producing Lachnidae (Hemiptera). In tropical primary forests, however, ants are always present as opportunistic predators, exerting high selection pressure on canopy arthropods. In both ecosystems, most arthropods were found on trees dominated by ants. While this may be an effect of honeydew abundance in temperate forests, it remains unexplained in tropical rainforests where honeydew-producing Hemiptera are not a reliable food resource. The comparison of arboreal ants with secondary forest ant communities reveals fundamental changes in diversity and structure associated with a reduction in predation pressure. This has a direct impact on arthropod fauna, some of which can become pests. The example of canopy arthropod communities shows that the regeneration of secondary forests takes more than 50 years—much longer than expected—and that forest regeneration is only successful when primary forests are available as a source for recolonization.

## Figures and Tables

**Figure 1 animals-15-01914-f001:**
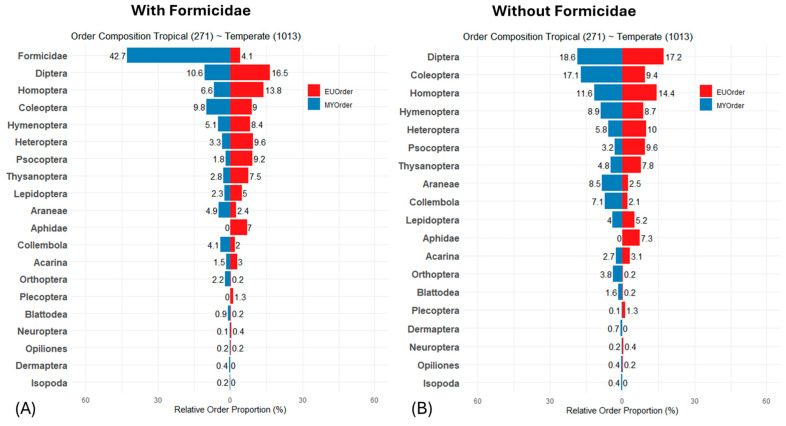
(**A**) The pyramid plot contrasts the proportional distribution of major arthropod groups fogged from different trees in tropical lowland forests (blue) with those from temperate forests of Central Europe (red). Aphids are visualized separately due to their high abundance in temperate forests and their absence in tropical forests. (**B**) An indication of how ants affect arthropod composition is provided by excluding ants from the proportional calculation. The number of foggings is indicated in brackets in the plot headings.

**Figure 2 animals-15-01914-f002:**
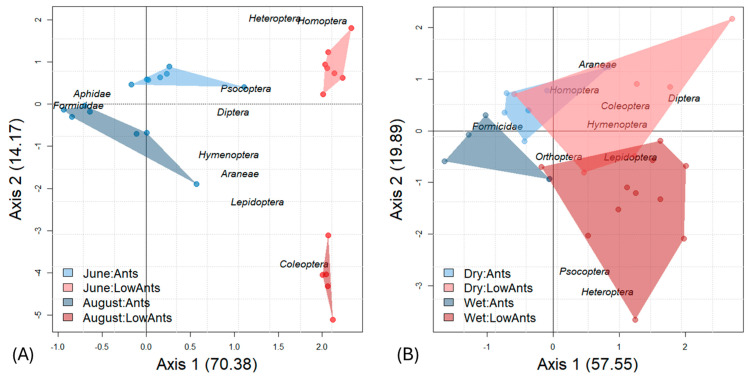
(**A**) A correspondence analysis distinguishes temperate arthropod communities dominated by *F. polyctena* ants from those without *F. polyctena* along the first axis, with the seasonal effect represented on the second axis. (**B**) These contrasts are also observed in lowland rainforests, although less pronounced.

**Figure 3 animals-15-01914-f003:**
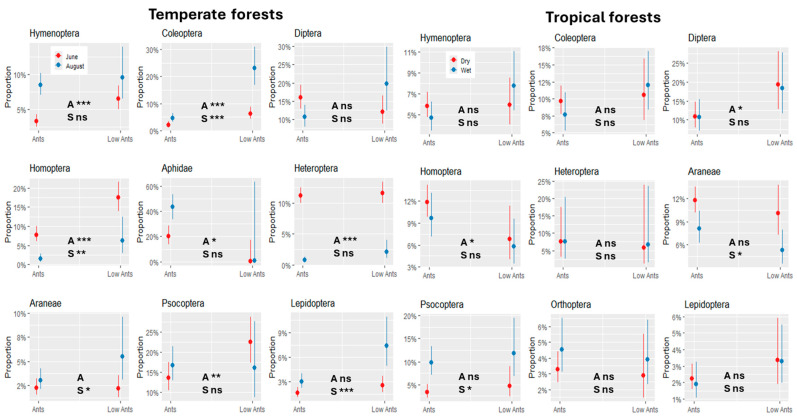
The marginal interaction plots of the logistic regression models show the proportional changes of the major arthropod orders on trees with high ant abundance (Ants) and low ant abundance (LowAnts) across different seasons in temperate forests and tropical forests. The interaction term Ants*Season was significant only for Diptera and Heteroptera in Europe. The significance of the ant impact (A) and the seasonal effect (S) are indicated in each plot. *p* < 0.05 (*), *p* < 0.01 (**), and *p* < 0.001 (***). ns: not significant.

**Figure 4 animals-15-01914-f004:**
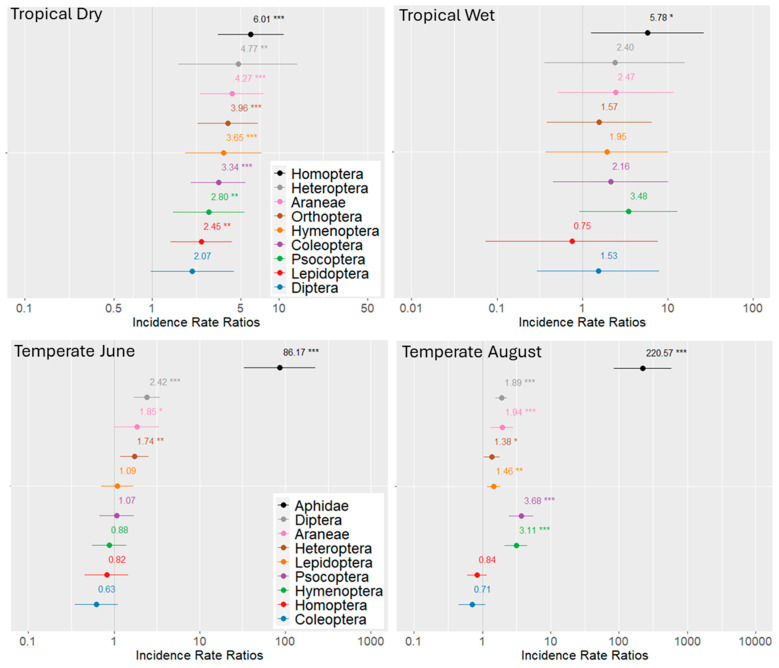
Forest plots of the negative binomial regression models illustrate the effects of ant dominance on arthropod abundance in tropical and temperate forests. The *x*-axis displays the incident rate ratios (IRRs) on a logarithmic scale, indicating the fold-change in abundance per unit increase in the predictor (ant presence). An IRR of 1 denotes no effect. In tropical forests, arthropod abundance on “Ant trees” increased predominantly during the dry season. Similar effects were observed in temperate forests in August. Points represent regression coefficients with 95% confidence intervals. Major arthropod orders are shown in distinct colors. Statistical significance is indicated by asterisks; coefficients not significantly different from 1 are plotted without asterisks. *p* < 0.05 (*), *p* < 0.01 (**), and *p* < 0.001 (***).

**Figure 5 animals-15-01914-f005:**
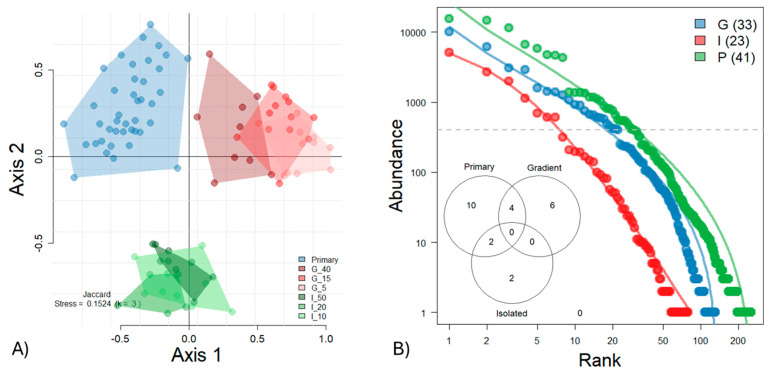
(**A**) NMDS ordination illustrates significant differences in Jaccard beta diversity among ant communities from primary forests (blue), gradient forests aged 5, 15, and 40 years (red), and isolated forests aged 10, 20, and 50 years (green) (see Appendix A). A clear trajectory toward primary forest conditions is observed only in the gradient forests. Forest types are abbreviated as P = Primary, I = Isolated, and G = Gradient, with numbers indicating forest age. (**B**) Ranked species abundance distributions reveal marked differences in ant community structure across forest types, with isolated forests exhibiting the least pronounced dominance hierarchy. The Venn diagram indicates limited species overlap among forest types, based on the more abundant species (*n* > 400 individuals).

**Table 1 animals-15-01914-t001:** Relative proportions and count data of the major arthropod groups (with margins) show compositional differences between trees with and without ant dominance in tropical (**A**) and temperate forests (**B**). Strong seasonal differences were found in temperate forests, while seasonal effects were less pronounced in the tropics. The number of foggings for each experiment is provided in brackets below.

**Formicidae**	25%	46.30%	30.10%	57.80%		**Formicidae**	1.10%	37.10%	0.40%	66.50%	
	*2429*	*26,937*	*3915*	*20,080*	** *53,361* **		*256*	*23,800*	*28*	*63,833*	** *87,917* **
**Aphidae**	0%	0%	0%	0%		**Aphidae**	0.40%	12.60%	0.70%	14.50%	
	*0*	*0*	*0*	*0*	** *0* **		*88*	*8107*	*45*	*13,896*	** *22,136* **
**Diptera**	17.50%	7.40%	18.40%	5.20%		**Diptera**	12%	10%	19.70%	3.50%	
	*1705*	*4298*	*2391*	*1817*	** *10,211* **		*2673*	*6431*	*1285*	*3403*	** *13,792* **
**Coleoptera**	7.90%	5.20%	8.40%	3.20%		**Coleoptera**	6.20%	1.40%	23%	1.60%	
	*767*	*3016*	*1093*	*1122*	** *5998* **		*1379*	*887*	*1504*	*1494*	** *5264* **
**Homoptera**	5.10%	6.40%	4%	4.10%		**Homoptera**	17.20%	4.90%	6.20%	0.50%	
	*498*	*3705*	*524*	*1420*	** *6147* **		*3845*	*3140*	*402*	*470*	** *7857* **
**Heteroptera**	7.60%	7.10%	11.40%	4.80%		**Heteroptera**	11.50%	7%	2.10%	0.30%	
	*734*	*4152*	*1488*	*1657*	** *8031* **		*2565*	*4519*	*134*	*259*	** *7477* **
**Hymenoptera**	4.20%	2.90%	4.70%	1.90%		**Hymenoptera**	6.40%	2.10%	9.60%	2.80%	
	*404*	*1694*	*612*	*654*	** *3364* **		*1434*	*1323*	*624*	*2720*	** *6101* **
**Psocoptera**	3.10%	1.60%	6.20%	3.80%		**Psocoptera**	22.30%	8.60%	16%	5.60%	
	*298*	*902*	*810*	*1311*	** *3321* **		*4980*	*5515*	*1047*	*5389*	** *16,931* **
**Araneae**	7.60%	6.30%	3.70%	3.40%		**Araneae**	2.20%	1.40%	5.50%	1%	
	*736*	*3668*	*483*	*1185*	** *6072* **		*482*	*905*	*359*	*975*	** *2721* **
**Lepidoptera**	2.50%	1.20%	2.30%	0.80%		**Lepidoptera**	2.50%	1%	7.40%	1%	
	*245*	*703*	*300*	*276*	** *1524* **		*561*	*664*	*482*	*984*	** *2691* **
**Orthoptera**	2.20%	1.80%	2.70%	1.90%		**Orthoptera**	0.39%	0.16%	0.44%	0.03%	
	*211*	*1028*	*354*	*665*	** *2258* **		*87*	*101*	*29*	*24*	** *241* **
**Others**	17.40%	13.90%	7.90%	13%		**Others**	17.81%	13.75%	9.11%	2.65%	
	*1694*	*8065*	*1029*	*4529*	** *15,317* **		*3978*	*8834*	*595*	*2543*	** *15,950* **
**Sum**	**9721**	**58,168**	**12,999**	**34,716**	**115,604**	**Sum**	**22,328**	**64,226**	**6534**	**95,990**	**189,078**
**(A)**	**Dry**		**Wet**		**Sum**	**(B)**	**June**		**August**		**Sum**
	**LowAnts (7)**	**Ants (9)**	**LowAnts (11)**	**Ants (5)**			**LowAnts (7)**	**Ants (7)**	**LowAnts (5)**	**Ants (7)**	

## Data Availability

The data presented in this study can be requested from the corresponding author.

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
