# Peer review of "Impact of Ants on the Order Composition of Canopy Arthropod Communities in Temperate and Tropical Forests"

_animals, 2025, doi:10.3390/ani15131914_

Round 1

Reviewer 1 Report

Comments and Suggestions for Authors

The authors have presented a fine paper that provides new insights on how ants function in relation to Arthropod diversity and density in both tropical and temperate forest canopies. I can't find much apart from e few comments below.

General comments

When you talk about abundance, which abundance are you talking about? Species abundance (i.e. L317), abundance of orders or abundance of individuals? It is not always clear.

The paragraph starting at L355 is really interesting. That aphids become abundant in the fall and produce more honeydew is well known. The ants profit on the decline in aphid predators. Maybe you should add a point about that.

From L367 onwards, please consider the following as possibly additional explanations. 1) Ant activity creates microhabitats in the trees increasing habitat complexity and more niches for other arthropods. 2) Positive feedback between predator and prey as higher concentration of food resources attracts ants and other predators, scavengers and parasitoids. 3) Ant dominance can exclude other predators.

In the final chapter (L385+), it is clear to me that you are discussing the tropical forests but it may not be clear to everyone. Please specify.

Specific comments

Chapter 2.1, first paragraph: Please add some details on the trees selected. How many of the seven trees had ants and no ants and were the same trees selected in August? You say newly selected trees so I should assume that the answer is no. However, how can you then be sure that they were not affected by the June sampling, i.e. how far apart were they?

In Fig S1 the number of trees is in parentheses and sums to 26 for the temperate region while in the text you say 7 trees and twice. Please clarify.

Figure 1: The underlying information from this figure can be confusing. The number of foggings is much higher than mentioned in the Materials and Methods chapter. Should I then assume that you have pooled from other sources? Please explain and add references if so. Also, what do the proportions represent? Biomass, number of species, individual counts? Finally, the plot heading should be in the legend to the figure as it is partly in German.

Figure 2: I guess that the purpose of the CA is to visualize the distinctions only. Otherwise, there is an arch effect in 2A.

Figure S3: It says 999 permutations. In the text (L185) you say 9999 permutations. Please clarify.

L344+ …approximately 30kg per....: per what? Insect, mound, total, canopy? Please specify.

Other comments

L83: Delete \

L139: Please consider to change “…substances in the trees” with “…substances in the environment”

Legend S2: A typo “tremperate” 2nd last line

L237: Typo “Juni”

L315: Reference should be to S-Fig 3?

Author Response

Dear reviewer,

Thank you for the helpful comments that clarified and improved the manuscript.

We have addressed all of the comments and incorporated most of the suggestions in the revised manuscript.

Best regards,

For both authors

Andreas Floren

#----------------------------------

Reviewer 2 Report

Comments and Suggestions for Authors

The manuscript entitled “Impact of ants on order composition of canopy arthropod communities in temperate and tropical forests” by Andreas Floren and Tobias Müller, submitted to the Animals magazine, presents data based on arthropod abundances in trees. Authors used a fogging technique, and presents results based on studies in temperate forests, as well as, tropical primary forests. The presented data are interesting, especially as composition of arthropods from forests in different regions were analysed. The data presented are worth to be published; however, I think that the manuscript could be improved. I hope that the following comments can be used to improve it.

General comments
Authors studied arthropod abundances. I believe that the found differences are result of the ant presence/absence; however, presently the direct influence was no studied, it is just interpretation of results. Thus, I think that the title (i.e. the part “Impact of ants on…”), as well as several part of the manuscript should be changed.

The Abstract section should be improved, as – for example – there is lack of information about the design of the study.
The goals of the study could be written in a better way. In the present form it is difficult to understand, what was the main goal of this study.

Authors used advanced statistical methods. I think that the analyses are correct; however, for statistical analyses the data should be independent. But see, for example, (lines 96-97) “seven fully grown oak trees—with and without F. polyctena—were fogged in each forest”, and (line 88) “The two forests were located one kilometer apart”. Thus, the data for the seven trees (from one and from another forest) are not independent, I think. I understand the problem of such study: it is rather impossible to find, e.g., 14 similar forests, and to collect data from one (preferably: randomly selected) tree in each of the forests. Nevertheless, the problem of (lack of) independence is important. Thus, at least, a short discussion of the subject in the Discussion section is recommended.

It is difficult to follow the Materials and methods and the Results sections, concerning the sample sizes. For example (lines 96-97): “In June, seven fully grown oak trees—with and without F. polyctena—were fogged in each forest”, line 113 “These included 14 trees” and line 121 “resulting in a total of 41 trees”, but line 196 “a pyramid plot based on the analysis of 1284 fogged trees” [thus, in the Materials and methods section, the mentioned samples are rather small, but in the Results – really large; additionally, in the Results section it is difficult to find the information on the used sample size, for example see data presented in Table 1, Figure 3, Figure 4]. I believe that everything is correct (see e.g. lines 132-133 “From each isolated forest, 9, 6, and 8 trees were fogged.” Thus, 9 multiplied by number of the forests +6 multiplied by number of the forests multiplied by number of the forests, etc.), but – again – it is difficult to follow it. Such information (on the sample sizes for the specific result) in the Results section should be clearly presented. Thus, consider to add the used sample sizes, for example, to the legend of tables and figures.
To sum up: the structure of the two section could be improved (concerning the sample sizes). Additionally, it should be easy to find, which part of Materials and methods section refers to which part of the Results section – in the present version it is difficult to find a relevant part of Materials and methods section regarding to the specific result (from the Results section).

Several specific comments
I am not sure, but it is not the best to call the trees with fewer than 500 individuals as “NoAnt-trees” – 400-500 individuals still is quite large number, I think. So, consider to change the name.

Additionally, give the information on the exact number of individuals (for both groups, i.e. the “Ant-trees” and the “NoAnt-trees”; for example, min-max values, and medians values).

Figure 1. The red colour group is on the right, and the blue colour group on the left. Please forgive me for my possible misunderstanding, but the legend “Order Composition Temperat (1013) ~ Tropen (271)” should be opposite I think, i.e. “Order Composition Tropen (271) ~ Temperat (1013)”.

Author Response

Dear reviewer,

Thank you for the helpful comments that clarified and improved the manuscript.

We have addressed all of the comments and incorporated most of the suggestions in the revised manuscript.

Best regards,

For both authors

Andreas Floren
